# Experimental and Analytical Studies of Prefabricated Composite Steel Shear Walls under Low Reversed Cyclic Loads

**DOI:** 10.3390/ma15165737

**Published:** 2022-08-19

**Authors:** Shenggang Chen, Xiaotong Peng, Chen Lin, Yingying Zhang, Hexiang Hu, Zhengjian He

**Affiliations:** 1Jiangsu Key Laboratory Environmental Impact and Structural Safety in Engineering, China University of Mining and Technology, Xuzhou 211116, China; 2School of Civil Engineering and Architecture, University of Jinan, Ji’nan 250022, China; 3School of Architecture and Landscape Design, Shandong University of Art & Design, Ji’nan 250014, China; 4China Construction First Group Corporation Limited, Beijing 100161, China; 5China Architecture Design & Research Group, Beijing 100044, China

**Keywords:** composite steel shear wall, prefabricated RC panels, seismic performance, boundary condition, predicting equation

## Abstract

Prefabricated composite shear walls (PCSW) consisting of steel plate clapped by single-sided or double-sided prefabricated reinforced concrete (RC) panels have enormous advantages for application as lateral-resisting structures in prefabricated high-rising residential buildings. In this paper, three 1/3-scaled PCSW were manufactured and tested to investigate the seismic performance of PCSW with single-sided or double-sided prefabricated RC panels. The experimental results, including hysteretic and skeleton curves, stiffness and strength degradation, ductility, energy dissipation capability and steel frame effects, were interpreted, compared and summarized. In spite of the RC panels being the same thickness, PCSW with double-sided RC panels had the most outstanding lateral-resisting properties: the highest yield strength and bearing capacity, adequate ductility, plumper and stable hysteresis loop and excellent energy absorption capacity. Finally, a simple predicting equation with a modification coefficient to calculate the effects of boundary steel frame was summarized and proposed to calculate the lateral yield load of the PCSW. All efforts were made to give reliable technical references for the design and construction of the PCSW.

## 1. Introduction

Steel plate shear walls (SPSW), i.e., thin steel plate walls used to resist lateral loads, have been demonstrated to be an effective structural system with high energy dissipation capacity and lateral load-resisting capacity [1]. Lateral stiffness of a steel plate shear wall can be analyzed by the diagonal compression field theory, which is greatly influenced by the boundary of steel plates [2]. Although SPSW can survive extensive inelastic deformation, great out-of-plane deformation may lead to a drastic drop in bearing capacity and structural stiffness in the later loading stage. Moreover, problems of loud noise and thermal insulation arise when SPSW are in service. Therefore, several improved projects [3,4,5,6] have been proposed to solve premature out-of-plane buckling failure and problems in the application period.

As shown in Figure 1a, the first improved version of SPSW involved the addition of stiffeners in horizontal, vertical or diagonal directions [7]. However, new troubles arise: (1) restriction of space for doors and windows; (2) uneconomical consumption of steel. The second version is the composite steel shear wall. Three main kinds of composite steel shear walls are summarized here. (1) The first form is composed of steel plates embedded in the reinforced concrete [8,9], as shown in Figure 1b. Generally, steel studs are used to guarantee the bond between steel plate and concrete. However, premature cracking of concrete greatly limits the application of such structures. (2) The second form is the internal concrete wall confined by external steel plates [4,10,11,12], as shown in Figure 1c. Similarly, shear studs and even stiffener plates are used for shear connectors. Many experimental and theoretical studies have been conducted. Perfect shear resistance capability and high ductility were realized in lab and field practice. However, the on-site work of wet operations cannot be avoided. (3) The third form, consisting of steel plate clapped by single-sided or double-sided prefabricated concrete panels, also called prefabricated composite shear wall (PCSW), is shown in Figure 1d. Convenient connections between steel plate and concrete panels are achieved by high-strength bolts, avoiding time-wasting and environmentally damaging wet operations [13,14,15]. Besides the improvement in the capacity to resist seismic loads, the fire-resistant and sound insulation performance are greatly improved as well.

Based on these benefits, PCSW have attracted much attention, especially due to the fact that Chinese government has been committed to vigorously promoting the application of prefabricated steel structures in high-rise residential buildings. PCSW will have more extensive application prospects [3]. PCSW were firstly promoted by Astaneh and Zhao [15,16], who designed and loaded traditional PCSW were. A gap between concrete panels and boundary frame was recommended in these studies. Guo et al. [17] were the pioneers who started the research into PCSW in China. A new version of PCSW was developed that enlarged the diameter of bolt holes in the concrete panels. Subsequently, a few studies researched the effects of different factors on the performance of PCSW, using both experimental methods and finite element analysis methods. These factors included stiffness of the concrete panels [18,19,20,21], distribution of bolts [14], perforated steel plates [1,22,23], steel plates with inclined slots [6,24,25] and partially connection types between frame and steel plate [19,26,27]. In the aforementioned studies, steel plate shear walls constrained by steel frames with hinge joints [17] or rigid connections [13,15] were studied. However, frames with semi-rigid connection are the widely applied type in practice. Few studies focus on the seismic behaviors of PCSW restrained by frames with semi-rigid connections. Furthermore, no studies were conducted to compare the cyclic performances of steel plate shear walls with single or double concrete panels. 

In this paper, three 1/3-scaled specimens, including SPSW and PCSW with single-sided and double-sided RC panels, were fabricated and loaded. The uniform semi-rigid steel frames were used to fix the shear walls. The performances of all the specimens, including with regard to their hysteretic and skeleton curves, stiffness and strength degradation, ductility, energy dissipation capability and steel frame effects were summarized, interpreted and compared. Furthermore, a simple predicting equation with a modification coefficient α to consider the effects of boundary steel frame was summarized and proposed to calculate the lateral yield load of PCSW.

## 2. Experimental Information

### 2.1. Design of Specimens

Referring to a real building structure, three specimens with 1:3 reduced scale were designed according to JGJ99-2015 [28] and JGJ/T380 [29] in this paper. As exhibited in Table 1, three kinds of shear walls connected to the uniform steel frame were designed and manufactured: a steel plate shear wall (denoted by SPSW), a double-sided prefabricated composite shear wall (denoted by DPCSW) and a single-sided prefabricated composite shear wall (denoted by SPCSW).

Details of the unified steel frame are displayed in Figure 2. The frame had two stories and one span. Shaped steel with I-section was used for the frame beams with a clear span of 1.575 m, whereas Hsection steel was applied for frame columns with a clear height of 1.11 m. Connections between steel beams and steel columns were semi-rigid joints accomplished by extended end-plates and bearing-type high-strength bolts. The depth of the end-plates was 9 mm. A schematic diagram of the beam–column connection is shown in Figure 3. The bolt diameter and spacing were designed with the principle that the bending moment caused by the tensile stresses of bolts should exceed the design bending moment of the semi-rigid connection.

Uniform steel plates were adopted by all specimens, as shown in Figure 4. The thickness of the steel plate was 2 mm, which satisfied the requirement of height-thickness ratio λ≤600 [29]. λ was defined by Equation (1) [29]. The connection of steel plate and steel frame was achieved by fish plates (angle steel 50 × 5) through welding. The fillet weld size was 6 mm.
(1)$$\lambda =H_{e}/t_{w}$$
where $H_{e}$ is the clear height of a story; and $t_{w}$ is the thickness of the steel plate.

PCSW, i.e., specimens of DPCSW and SPCSW, comprised two parts: steel plate and prefabricated RC panels. Concrete panels reinforced by A6@200 were prefabricated and attached to the surface of the steel plate by evenly distributed bolts. As shown in Figure 4, the inner steel plate was clamped by double-sided RC panels in DPCSW. As for SPCSW, RC panels only existed on one side of the steel plate. Based on the recommendation provisions in the Chinese code JGJ/T 380-2015 [29], the thickness of the RC panels was appropriately designed to prevent the out-of-plane global buckling. The thickness of the RC panel in the SPCSW was 70 mm, which equaled the total thickness of the two panels in the DPCSW. Bolt connections were applied to achieve reliable connection between RC panels and steel plate. There were in total 24 bolts with a distribution of 4 rows × 6 columns (as shown in Figure 4). To mitigate the crushing effect due to the deformable frame, structural joints were designed between the RC panels and the boundary frame. According to the Chinese code JGJ/T 380-2015 [29], the minimum width of the structural joints should satisfy the following Equation (2):(2)$$\Delta =H_{e}[\theta_{p}]$$
where $\theta_{p}$ is the elastic plastic story drift angle whose allowable value is 1/50 according to Chinese code JGJ 99-2015 [28]. It was calculated that the width of structural joints in the DPCSW and SPCSW could be taken as 22 mm.

### 2.2. Properties of Material

Three sets of 150 × 150 × 150 mm^3^ concrete cube blocks were cast and tested to obtain the cubic compressive strength *f*_cu_. Based on the test results, the average value of *f*_cu_ reached 42.08 MPa. Therefore, the prism compressive strength of the concrete can be defined as 0.76 *f*_cu,_ according to the Chinese code for the design of concrete structures GB50010-2010 [30].

The steel grade of Q345 was selected according to the specifications in JGJ99 [28] and JGJ/T380 [29]. The material properties of the steel frame and steel plates were also tested. Detailed information on the yield strength, ultimate strength and elastic modulus is recorded in Table 2.

### 2.3. Test Setup

The loading setup is composed of three components: (1) the lateral loading device; (2) the supporting system and (3) the boundary system. As shown in Figure 5, horizontal cyclic loads were achieved by the MTS actuators with measuring range of 100 t. Generally, a load transfer beam was used. A supporting system located on the top and middle frame beams was specially designed to prevent out-of-plane buckling instability in the frame. Support rollers made the frame deform freely in the plane direction whereas they limited its out-of-plane deformation. The boundary system, which included ground beam and high-strength bolts, guaranteed the fixed condition of steel frame.

### 2.4. Arrangement of Measuring Points 

During the loading procedure, a series of experimental data were measured and transmitted into the controlling computer by the IMP board. Three kinds of results were collected: force, displacements and strains. Horizontal force was automatically recorded by the force sensor. The locations of the displacement meters and strain gauges are displayed in Figure 6. The lateral shifts of each story were monitored by DL1–DL3, whereas shear deformation of the shear walls and joints were monitored by DS1–DS4 and DS5–DS6. Further, readings of DB1–DB4 exhibited bending deformation of the joints. The stresses of the frame columns, frame beams and shear walls were measured by strain rosettes SC1–SC8, SB1–SB2 and SW1–SW12, respectively.

### 2.5. Testing Procedures

The cyclic loads were achieved by the displacement loading control method. It was assumed that the direction of the displacement was positive with the push of the actuator and negative with pull of the actuator. As shown in Figure 7, the whole loading process included two procedures: the preloading procedure and the formal loading procedure. In the preloading procedure, a lateral displacement of ±5 mm was loaded and unloaded to eliminate the initial clearance and to guarantee all the instruments were in normal working condition. Subsequently, the formal loading procedure was implemented. The displacement increment was set to 3 mm and three loading cycles were conducted at each loading level. Finally, the loading process was terminated either by the collapse of the structure or when the lateral forces dropped below 85% of the drift of maximum strength. 

## 3. Experimental Results and Discussion

### 3.1. General Performance

For a better understanding, $\Delta_{y}$ was taken as the basic unit of loading displacement. $\Delta_{y}$ was defined by lateral loading displacement corresponding to the yield point, which can be calculated by the graphic method [31] illustrated in Figure 8. The values of $\Delta_{y}$ from the different specimens can be found in Table 3. Firstly, specimen SPSW was taken as an example to illustrate the damage process:
(1)Before the loading displacement reached $\Delta_{y}$, no obvious changes were found aside from the squeaking due to the friction between gaskets and shear walls. The deformation of the specimen was able to be completely recovered under cyclic loads. (2)When the loading displacement reached $\Delta_{y}$, the lateral force was $0.84F_{p}$, where $F_{p}$ is the peak value of the lateral loading force. Figure 9a reveals that slight local buckling was observed in the steel plate shear walls. Tension struts in the direction of the principle tensile stress were mainly located in the diagonals. It is noted that the range of local buckling in the second story was larger than that in the first story. When the specimen was unloaded to zero, slight residual deformation could be detected. (3)When the loading displacement reached $1.9\Delta_{y}$, the lateral force reached $F_{p}$. During the loading process, local buckling become more evident (as shown in Figure 9b), as demonstrated by the following two aspects: ① the numbers of tension diagonal struts increased; ② the range of the tension diagonal struts became larger. During the unloading process, the recovery of buckling deformation was very slow. When the specimen was unloaded to zero, obvious residual deformation in the shape of an X was still visible. (4)When the loading displacement reached $2.4\Delta_{y}$ the structure entered the failure stage. The experiment was stopped as the lateral loading force decreased below 85% of $F_{p}$.

In the loading process, specimens of SPCSW and DPCSW exhibited some unique characteristics because of the addition of concrete panels. The constraints provided by concrete panels were able to effectively enhance the compression capacity of the steel plate, avoiding premature buckling failure. After the completion of the experiments, steel plates and concrete panels were separated from each other so that their respective performances could be observed and analyzed carefully. 

The steel plates of all three specimens are shown in Figure 10a as they were when the structures were in ultimate bearing condition. It was revealed that different buckling conditions appeared. Compared with specimen SPSW, more tension diagonal struts formed in specimen SPCSW. The application of the RC panel, working in the stiffening manner of ribs to provide out-of-plane constraint, enabled the principle compressive stress to increase while avoiding the earlier buckling failure of the compressive yielding of the steel plate. Consequently, the failure mode of SPCSW changed from low-order buckling to high-order buckling. A much stronger constraint could be achieved by the double-sided concrete panels in specimen DPCSW. The constraint was so strong that there was no difference in the tensile and compressed mechanical performances of the steel plate, which made the central region enclosed by the bolts behave as thick plates without out-of-plane deformation. Therefore, the failure mode of the DPCSW was the shear yield of steel plate. As shown in Figure 10a, dense local buckling was found in the corners of the steel plate because of the weaker constraints of the free edges. Similar characteristics were found in the morphology of the RC panels from specimens SPCSW and DPCSW, shown in Figure 10b. Gridded gray impressions formed around the panel, which was due to the extrusion between the RC panels and tension diagonal struts. However, almost no impression occurred in the central region of the panel because of the stronger constraints originating from bolts.

Moreover, it was noted that no cracks were observed in the RC panels before the yield of the structures. Large-area cracks only appeared when the structure was at the later loading stage. The concrete panel of specimen SPCSW remained nearly intact, whereas the edges of the concrete panels of specimen DPCSW were seriously damaged. This was because the free edges of the concrete panels were less constrained and the structural gap was too small.

### 3.2. Force-Displacement Response

#### 3.2.1. Hysteretic Curves

Hysteretic curves related to lateral force (denoted by *p*) and overall lateral displacement (denoted by Δ) are drawn in Figure 11. Generally, the following conclusions could be drawn: (1) Comparisons among specimens SPSW, SPCSW and DPCSW indicated that the introduction of RC panels converted the shapes of the hysteretic curves from a reverse S-shape with pivot pinching points to a rounded shape with a slight pinch. The areas of the hysteretic loops from specimen DPCSW were much larger than those from specimens SPSW and SPCSW. (2) In each loading cycle, lateral displacement increased with the increase in lateral force, whereas structural stiffness (slope of curves) decreased. The maximum loading force gradually decreased and lateral stiffness reduction accelerated in different cycles at the same loading level. Obviously, the degree of decrease in DPCSW was the smallest. (3) Results showed that the addition of RC panels could effectively restrain the buckling of steel plates and induce multiple modes of energy dissipation in the structure. Furthermore, there was a superior effect in the shear walls with double-sided concrete panels than in those with single-sided panels. 

#### 3.2.2. Skeleton Curves

For better understanding of hysteretic performance of specimens, the skeleton curves shown in Figure 12 were converted from hysteretic curves. Meanwhile, details of characteristic points are summarized in Table 3. Similar development stages of the mechanical properties were observed: (1)The elastic stage, where a linear relationship between the lateral drifts and forces occurred. The curves of all the specimens coincided with each other. Similar initial stiffnesses (slope of the curves) were observed, which meant the attached RC panels had little effect on the initial stiffness of the shear walls.(2)The plastic stage, which initiated from the yield point and ended at the ultimate strength point. The restraining effect of the concrete panels greatly increased the yield strength and maximum bearing capacity of the shear walls. Compared with specimen SPSW, the yield strength and ultimate strength of SPCSW increased by 9.34% (5.21%) and 25.96% (33.85%). The more noticeable growth in the yield strength and the maximum strength of specimen DPCSW were revealed as 40.30% (32.32%) and 85.59% (93.31%), respectively. Numbers outside and inside the parentheses represent values corresponding to the positive and negative loading directions, respectively.(3)The failure stage, where lateral forces descended along with the continuous increase in lateral drift.

### 3.3. Strength Degradation

The mechanisms of strength degradation can be revealed by the strength degradation ratio ($\xi_{i}$), which is defined by the ratio between the maximum strength in the *n*_th_ cycle and that in the first cycle at the same loading level *i*. $\xi_{i}$ is expressed by Equation (3).
(3)$$\xi_{i}=F_{i}^{n}/F_{i}^{1}$$
where *n* is the number of cycles at the loading level *i* and $F_{i}^{1}$ and $F_{i}^{n}$ are the first and *n*_th_ loading cycle, respectively, at the loading displacement level *i*. 

Figure 13 gives the changes in $\xi_{i}$ at different lateral loading levels. It indicates that $\xi_{i}$ gradually decreased with the increase in loading cycles at the same loading level. However, the reduction process of $\xi_{i}$ was clearly divided into two stages: (1) Before the approach of maximum strength, $\xi_{i}$ was barely decreasing. The values ranged from 0.97 to 1.00. (2) After the maximum strength point was reached, $\xi_{i}$ dramatically declined. The values of $\xi_{i}$ were less than 0.87, 0.72 and 0.90 for the three specimens, respectively. The higher value of $\xi_{i}$ indicates a favorable strength-maintenance capacity for specimen DPCSW.

### 3.4. Stiffness Degradation

The stiffness degradation of the specimens subjected to cyclic lateral force can be evaluated by the loop stiffness $K_{m}$ (kN/mm). The physical meaning of $K_{m}$ was the average secant stiffness of different loading cycles at the same loading level. $K_{m}$ is calculated by the Equation (4): (4)$$K_{m}=\sum_{i=1}^{n}P_{m}^{i}/\sum_{i=1}^{n}\Delta_{m}^{i}$$
where $P_{m}^{i}$ is the maximum strength in the *i*_th_ loading cycle at the *m*_th_ loading level; $\Delta_{m}^{i}$ is the lateral drift corresponding to $P_{m}^{i}$; and *n* is the total loading cycles at the *m*_th_ loading level.

The laws of stiffness degradation can easily be expressed by the degradation ratio β defined by Equation (5).
(5)$$\beta =K_{m}/K_{0}$$
where $K_{0}$ is the initial stiffness. In the elastic loading stage, the initial stiffness (i.e., tangential stiffness) was equal to the secant stiffness. Therefore, the initial stiffness could be calculated by the secant stiffness in the first loading level. 

Figure 14 displays variations in stiffness and the stiffness degradation ratio during the loading history. The initial stiffness of all three specimens had similar values. The steel plate and concrete panels had no interaction at lower loading level, which meant the concrete panels had no influence on the initial stiffness of tested specimens. As shown in Figure 14, the higher the loading level was, the lower lateral stiffness would be. When the loading level ranged from $-\Delta_{y}$ to $\Delta_{y}$, a sharp drop in stiffness happened in all three specimens. Subsequently, the stiffness degradation tended to be slower. Overall, the sequence of lateral stiffness values in the three specimens at the same loading level was: DPCSW > SPCSW > SPSW. The stiffness degradation was more stable and slower in specimen DPCSW than in SPCSW and SPSW. At the maximum strength point, the stiffness degradation ratios of all the specimens ranged from 0.25 to 0.30 in the positive loading direction, and from 0.29 to 0.31 in the negative loading direction. It should be noted that both the failure stiffness and the initial stiffness of all three specimens (as listed in Table 3) had only small differences, which indicated the addition of RC panels could not enhance the stiffness values. However, the degradation process of DPCSW was significantly slowed down.

### 3.5. Deformation Capacity–Ductility

Generally, structural deformation capacity was evaluated by the parameter of ductility μ, which is defined by the ratio of maximum lateral displacement ($\Delta_{m}$) to yield displacement ($\Delta_{y}$), as described in Equation (6). The results are recorded in Table 3 and illustrated in Figure 15. The fact that the ductility of all specimens exceeded 2.39 manifested in all three kinds of shear walls having a good deformation capacity. Comparisons among specimens SPSW, SPCSW and DPCSW showed that the RC panels could significantly increase the values of maximum displacement, although there were few changes in the yield displacement. Consequently, the ductility of specimens SPCSW and DPCSW increased by 79.5% and 34%, respectively, compared with specimen SPSW. RC panels can evidently postpone the overall buckling of steel plate, leading to a better deformation capacity.
(6)$$\mu =\Delta_{m}/\Delta_{y}$$

### 3.6. Energy Dissipation Capability

Energy dissipation capacity is an important aspect in evaluating seismic performance of structures. The energy dissipation coefficient $E_{e}$ and equivalent damping coefficient $\zeta_{e}$ are usually adopted to compute energy absorbed by a structure. High values of coefficients are favorable for structures to dissipate energy and to resist seismic loads. As illustrated in Figure 16, the values of $E_{e}$ and $\zeta_{e}$ in each hysteresis loop can be obtained by Equations (7) and (8):(7)$$E_{e}=S_{AEBF}/(S_{\Delta OAD}+S_{\Delta OBC})$$
(8)$$\zeta_{e}=E_{e}/(2\pi )$$
where $S_{AEBF}$ is the area surrounded by a complete hysteretic loop and $S_{\Delta OAD}$ and $S_{\Delta OBC}$ are the areas of the two triangles.

Figure 17 gives the relationship between the equivalent damping coefficient and lateral loading displacement. $\zeta_{e}$ increased when the loading displacement increased. When the loading displacement was less than $\Delta_{y}$, the coefficients remained at the lower values, from 0.05 to 0.1. When the loading displacement exceeded $\Delta_{y}$, the coefficient of specimen DPCSW grew much more quickly than other specimens. At the ultimate state, the value of the coefficient from DPCSW was over 0.26, whereas that of SPSW and SPCSW ranged from 0.12 to 0.15. For further clarification on the fact that double-sided attached RC panels greatly improved the energy dissipation capacity by enlarging the maximum drift, cumulative energy consumption $E_{sum}$ throughout the whole loading history was calculated, and is shown in Figure 18. The energy consumed by DPCSW was 14.09 times greater than SPSW, whereas the energy consumed by SPCSW was 1.8 times as much as SPSW. 

### 3.7. Distribution of Shear Resisted by Shear Walls and Steel Frames

Based on the measurement of stress conditions in the steel frames, lateral forces resisted by shear walls and frame columns were separated. As shown in Figure 19, the percentage of lateral forces resisted by shear walls against the total lateral forces in each story were summarized. For specimen SPSW, more than 80% of the total lateral forces were resisted by the shear wall, whereas the proportion was more than 90% for specimens SPCSW and DPCSW. Consequently, shear walls bore the vast majority of lateral forces. In design, it is safe to assume that the lateral forces were completely resisted by the shear walls. 

## 4. Prediction of Bearing Capacity

As illustrated in Figure 19, the structural frame has a certain influence on the lateral force resisted by composite shear walls. In order to analyze the interaction effects between boundary frame and shear wall, several SPSWs and CSSWs tested by the authors and other researchers [5,13,15,17,32,33] were collected and summarized. Details of the tested specimens are exhibited in Table 4.

According to Seismic Provisions for Structural Steel Buildings (AISC 341) [34], the boundary frame should be of sufficient stiffness to guarantee the effective bearing capacity of a shear wall. The minimum flexural stiffness of the boundary column must satisfy Equation (9) [34]. It is meaningless to compare the absolute value of each frame column in different experiments. Therefore, the effect of the boundary frame was normalized by introducing the parameter of η, which is the ratio between the actual stiffness of the frame column and the minimum stiffness that is required. With this normalization, results determined in different studies can be compared.
(9)$$I_{\min}=0.00307H^{4}t/L$$
where *H, t, L* are the height, thickness and span of the steel plate, respectively.

To evaluate the lateral yield load of shear walls mathematically, it is assumed that the shear yield point of steel plate can be achieved. The value of shear stress is expressed by $\tau =f_{y}/\sqrt{3}$, in which $f_{y}$ is the yield strength of steel plate. The following Equation (10) is used to predict the bearing capacity [29]. It is noted that coefficient α was added to modify the result to consider comprehensive influence of steel frames.
(10)$$P=\alpha \cdot f_{y}Lt/\sqrt{3}$$

All the calculated values of η and α were also given in Table 4. Figure 20 gives a more intuitive way to display the possible relationship between η and α. Analysis were conducted in two parts: 

(1) For SPSWs. As shown in Figure 20a, the lateral yield loads of SPSWs were indeed influenced by the stiffness ratio and the connection types of the frames. The values of α increased as the frame connection become stronger. The beam-connected-only type had the lowest value of α=0.345. The effect of the semi-rigid connection was similar to that of the simply supported connection. Meanwhile, there was almost no change in α as the value of η varied from 1.38 to 19.85. That is to say, the frames did little to strengthen the steel plate shear walls in this condition. The average value of the tested specimens was calculated as α=0.743. α<1 means the shear yield of the steel plate was not reached when the shear walls lost bearing capacity. In other words, buckling failure occurred. As for the SPSWs with rigid connected frames, a higher stiffness ratio in the frame column led to a higher yield load in the SPSW. If a linear relationship between α and η is assumed (red line in Figure 20a), parameter α can be derived by
(11)α=0.0362η+0.744

α of specimen 1 [32] with rigid connection was 0.47, which was significantly smaller than the other specimens with rigid connections. The reason for the lowest bearing capacity was the poor stiffness of the frame columns. η=0.75, being less than 1, indicates the required minimum stiffness was not satisfied.

(2) For DPCSWs. Similar phenomena appeared in composite shear walls. The simply supported and semi-rigid connections had similar effects on the yield load of DPCSW. The average value of α was 0.962, which was nearly equal to 1. Failure mode can be treated as the shear yielding failure of steel plate. The addition of prefabricated RC panels improved the lateral-resisting performance by preventing premature buckling failure. Hence, the parameter α of DPCSW was greater than that of SPSW. The stronger constraint of the rigid connection obtained higher value of α. Additionally, the greater stiffness of the frame columns also led to higher lateral yield load under the rigid connection condition. Similarly, the hypothesis of a liner relationship between η and α was adopted. Parameter α was calculated by the following Equation (12).
(12)α=0.0343η+1.368

To validate the correctness and application of Equation (10) for the bearing capacity of DPCSW, the formula was applied to analyze the ultimate shear stress of the tested specimens from Qi et al. [18] and Jin et al. [6]. Figure 21 exhibits the comparisons between the experimental results and the predicted results. It is evident that the predicted results coincided well with the experimental results. The determination coefficient R^2^ was 0.965 and mean absolute percent error (MAPE) was 6.06%. 

## 5. Conclusions

Three 1/3-scaled specimens, i.e., a steel plate shear wall (SPSW), a single-sided prefabricated composite shear wall (SPCSW) and a double-sided prefabricated composite shear wall (DPCSW), were manufactured and tested under low reversed cyclic loads to study the seismic performance of steel plate shear walls with single-sided or double-sided RC panels. Meanwhile, simple equations for predicting the lateral yield load were proposed to calculate the effects of boundary steel frames. The following conclusions could be drawn based on the results of the tested specimens:(1)Comparisons of the experimental results from the tested specimens SPSW and SPCSW demonstrated that SPCSW was a more effective choice to resist cyclic loads. Yield strength, ultimate bearing capacity, ductility and cumulative energy consumption of the SPCSW tested in this paper increased by 7.28%, 29.91%, 34.26% and 79.96%, respectively. However, the absence of an RC panel on one side of the steel plate meant that the premature buckling failure of the steel shear wall could not be fundamentally avoided.(2)With the same overall thickness of RC panels, the DPCSW tested in this paper displayed the most excellent lateral-resisting properties: the highest yield strength and bearing capacity, adequate ductility, plumper and stable hysteresis loop and excellent energy absorption capacity. The expected failure mode, i.e., shear yield failure of the steel plate occurring earlier than the buckling failure, was achieved.(3)Both the initial and failure stiffness of the three tested specimens were almost equal to each other. However, the addition of RC panels was able to effectively slow down the lateral stiffness degradation process. RC panels did not directly participate in resisting lateral loads. Therefore, almost no cracks appeared until the out-of-plane buckling deformation arose.(4)The effects of steel frames on the yield loads of specimens SPSW, SPCSW and DPCSW were considered by modification coefficient α. Steel frames with simply supported and semi-rigid connections had little influence on the lateral yield load of specimens as long as the minimum frame column stiffness was guaranteed. Nevertheless, the rigidly connected steel frame indeed elevated the lateral yield load. Furthermore, the higher the value of the stiffness ratio η of the frame columns, the greater the improvement degree.

## Figures and Tables

**Figure 1 materials-15-05737-f001:**
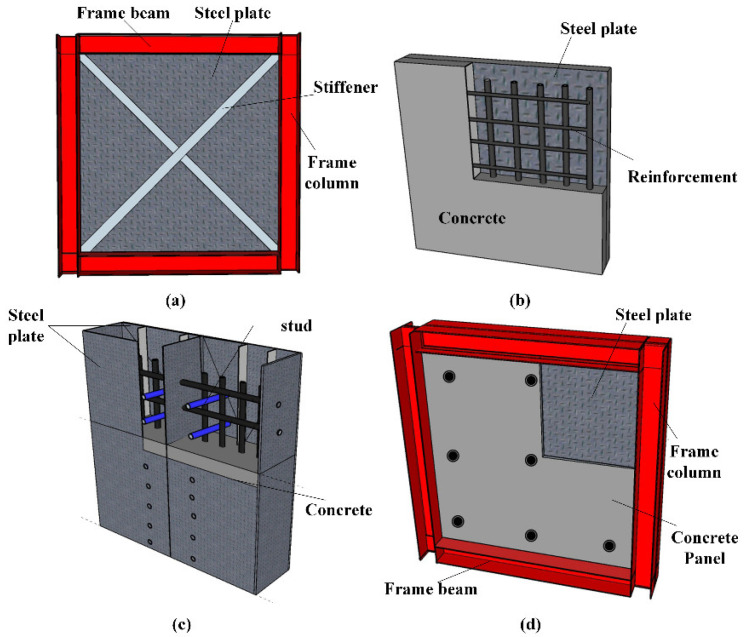
Different types of steel plate shear walls: (**a**) SPSW with stiffener; (**b**) steel plate embedded in concrete; (**c**) concrete wall confined by external steel plates; (**d**) PCSW.

**Figure 2 materials-15-05737-f002:**
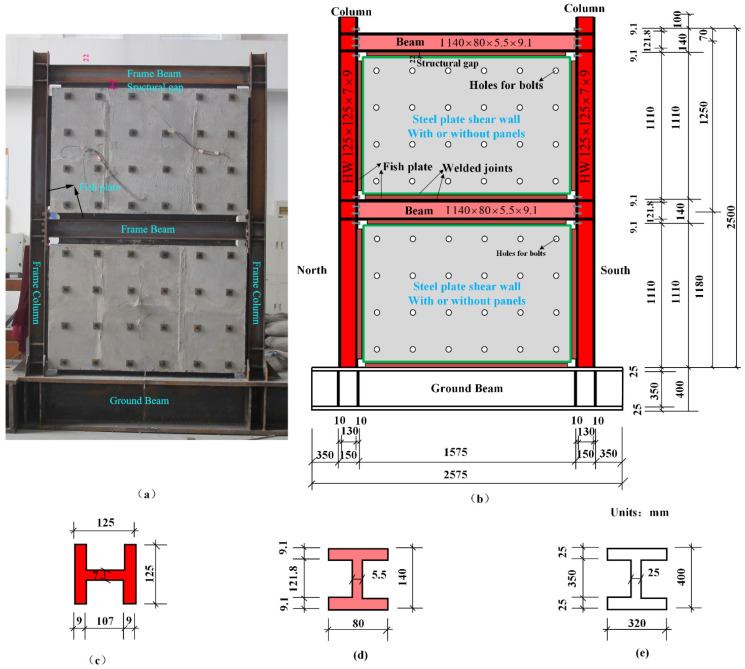
Dimensions of the steel frame: (**a**) tested specimen; (**b**) schematic diagram of tested specimen; (**c**) column; (**d**) beam; (**e**) ground beam.

**Figure 3 materials-15-05737-f003:**
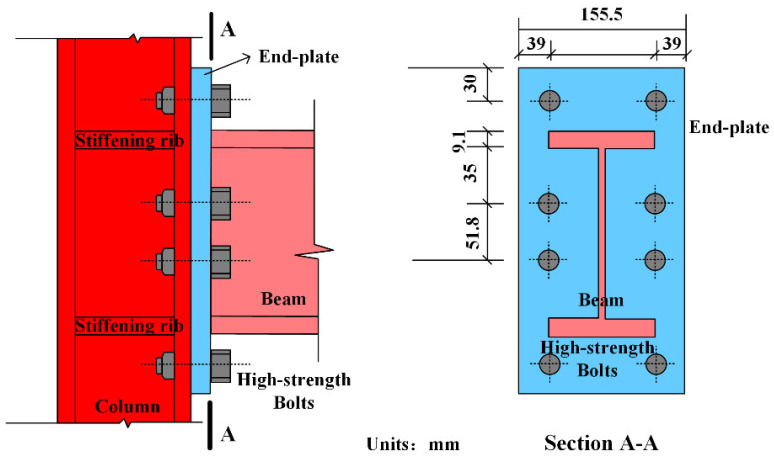
Schematic diagram of beam–column connection.

**Figure 4 materials-15-05737-f004:**
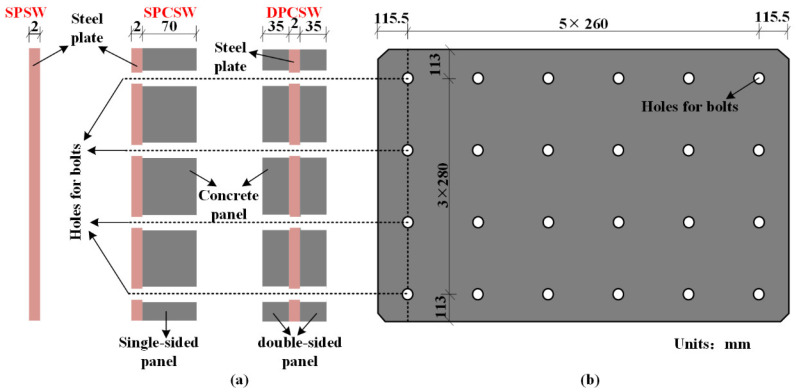
Composition of composite shear walls: (**a**) side view; (**b**) front view.

**Figure 5 materials-15-05737-f005:**
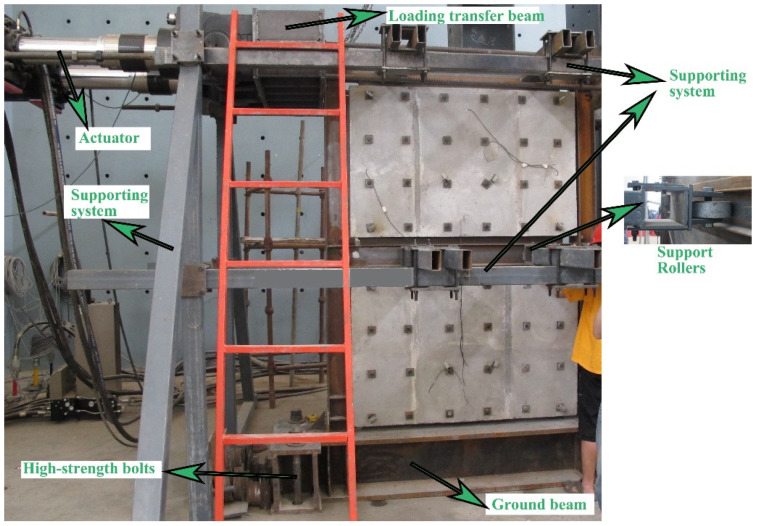
Test setup of a specimen of SPCSW.

**Figure 6 materials-15-05737-f006:**
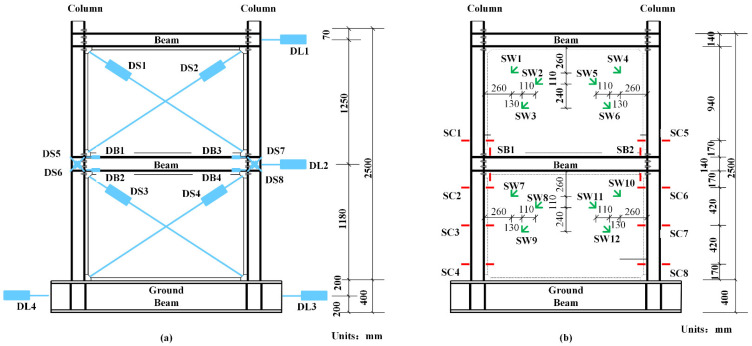
Plan of measuring points: (**a**) positions of displacement meters; (**b**) positions of strain gauges.

**Figure 7 materials-15-05737-f007:**
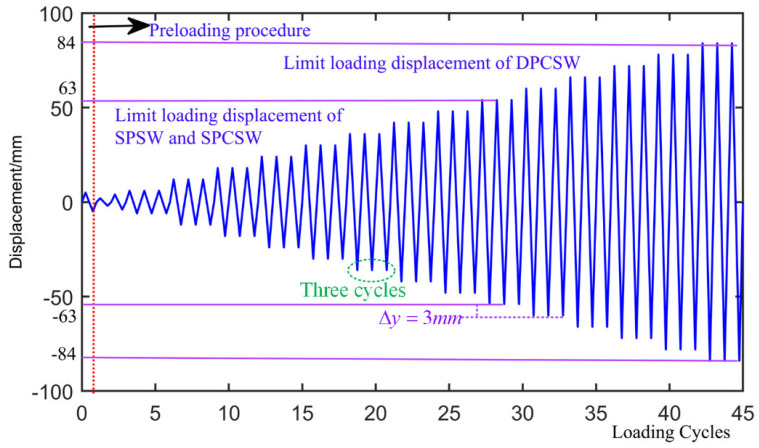
Loading systems of tested specimens.

**Figure 8 materials-15-05737-f008:**
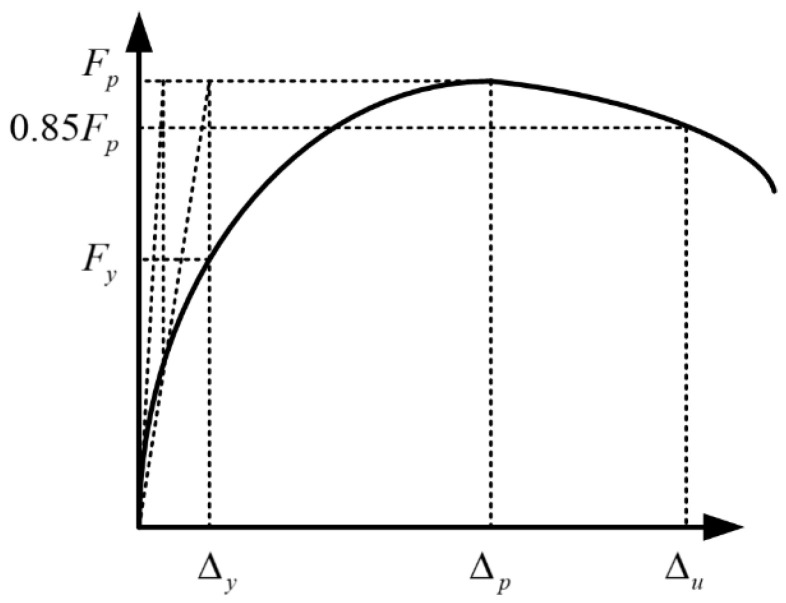
Graphic method to define yield point.

**Figure 9 materials-15-05737-f009:**
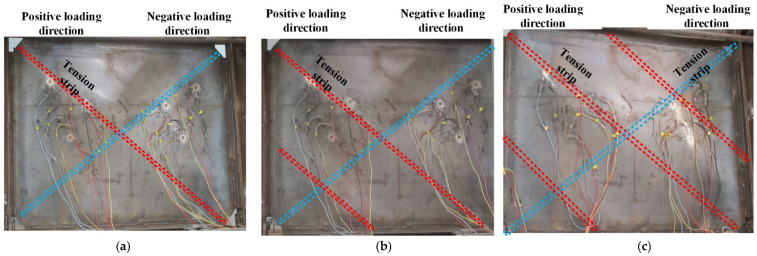
Performance of specimen SPSW: (**a**) loading displacement ∆*_y_*; (**b**) loading displacement 1.9∆*_y_*; (**c**) loading displacement 2.4∆*_y_*.

**Figure 10 materials-15-05737-f010:**
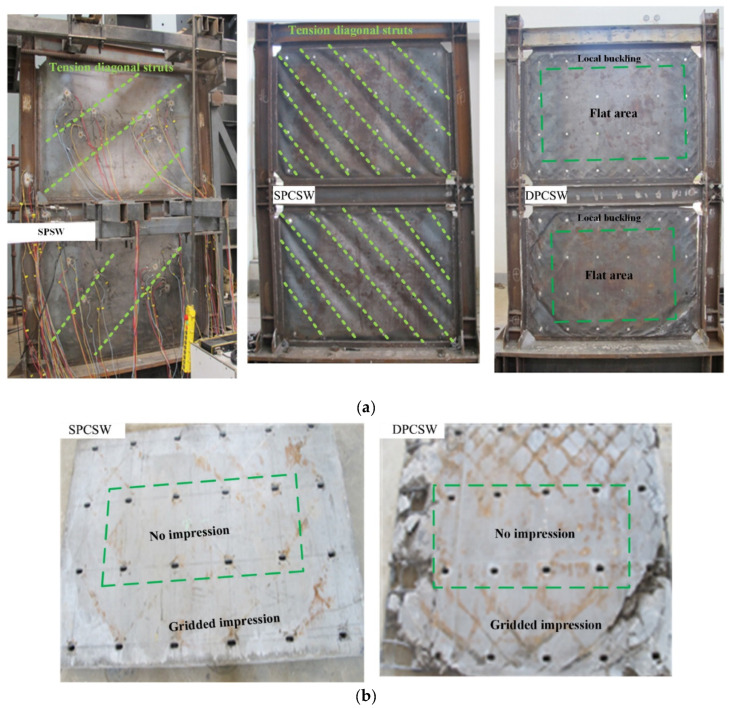
Failure mode of tested specimens: (**a**) performance of steel plates; (**b**) performance of prefabricated RC panels.

**Figure 11 materials-15-05737-f011:**
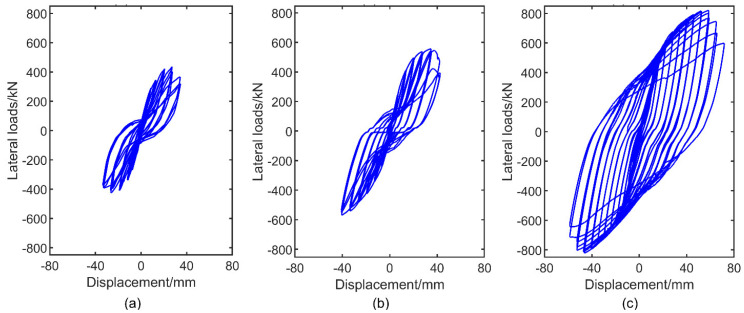
Hysteretic curves of specimens under displacement control loading: (**a**) SPSW; (**b**) SCSW; (**c**) DCSW.

**Figure 12 materials-15-05737-f012:**
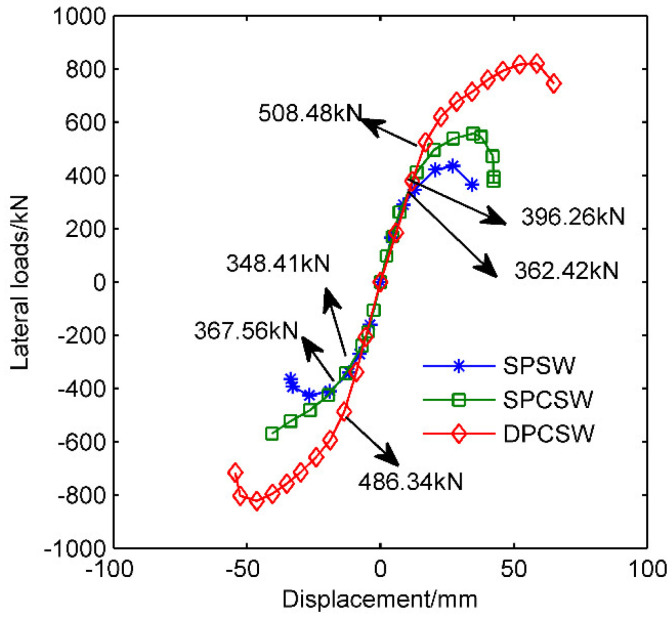
Skeleton curves of different specimens.

**Figure 13 materials-15-05737-f013:**
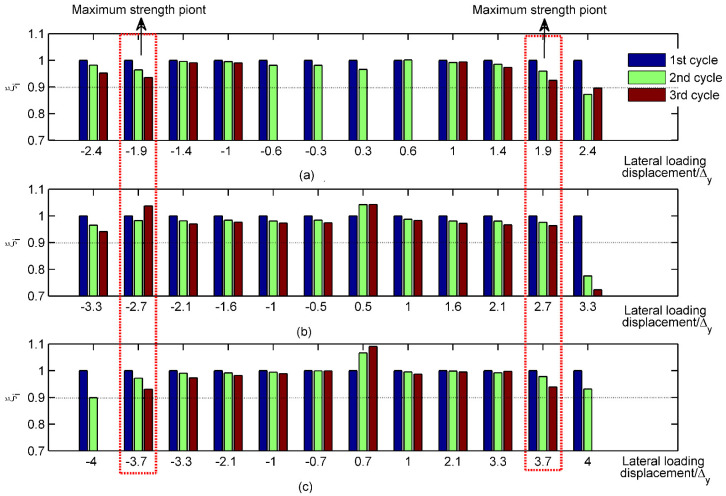
Strength degradation ratio of different specimens: (**a**) specimen SPSW; (**b**) specimen SPCSW; (**c**) specimen DPCSW.

**Figure 14 materials-15-05737-f014:**
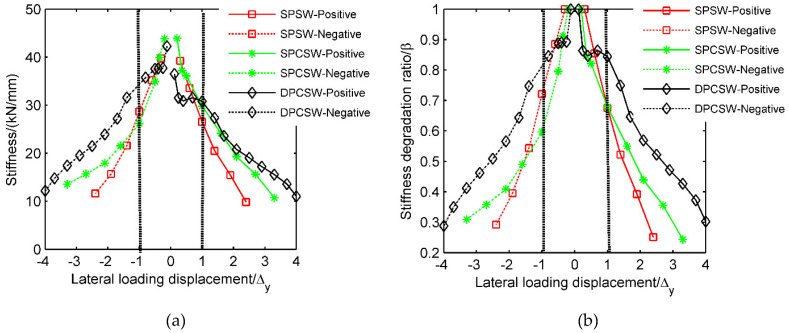
Stiffness degradation of three specimens: (**a**) changes of stiffness; (**b**) changes of stiffness degradation ratio.

**Figure 15 materials-15-05737-f015:**
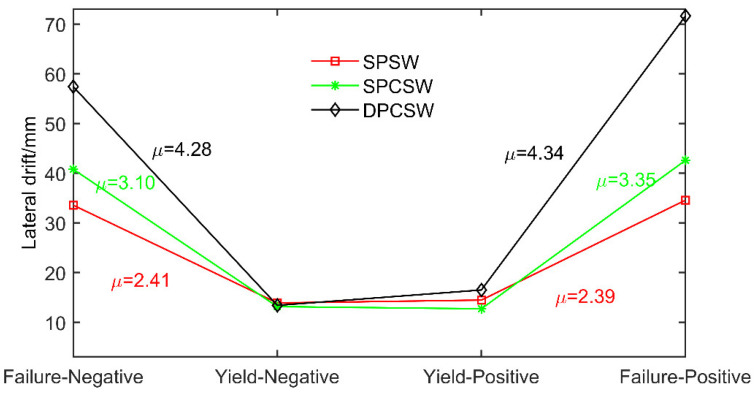
Ductility of all three specimens.

**Figure 16 materials-15-05737-f016:**
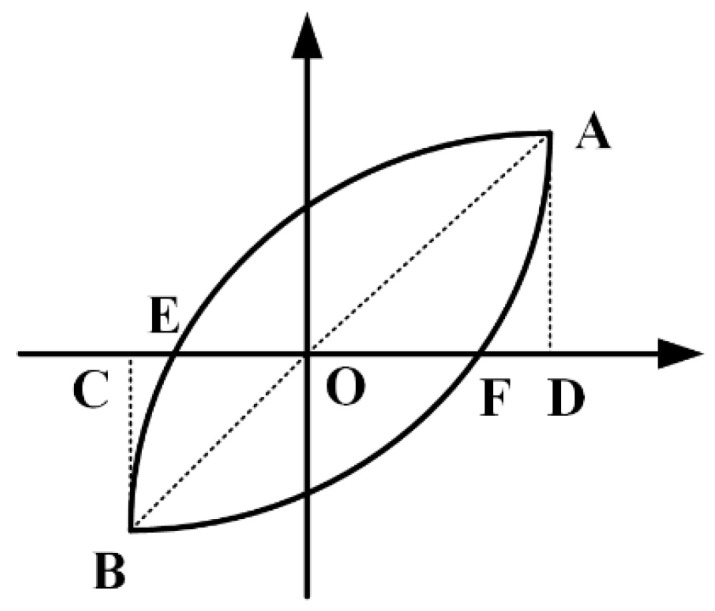
Schematic diagram of $E_{e}$ and $\zeta_{e}$.

**Figure 17 materials-15-05737-f017:**
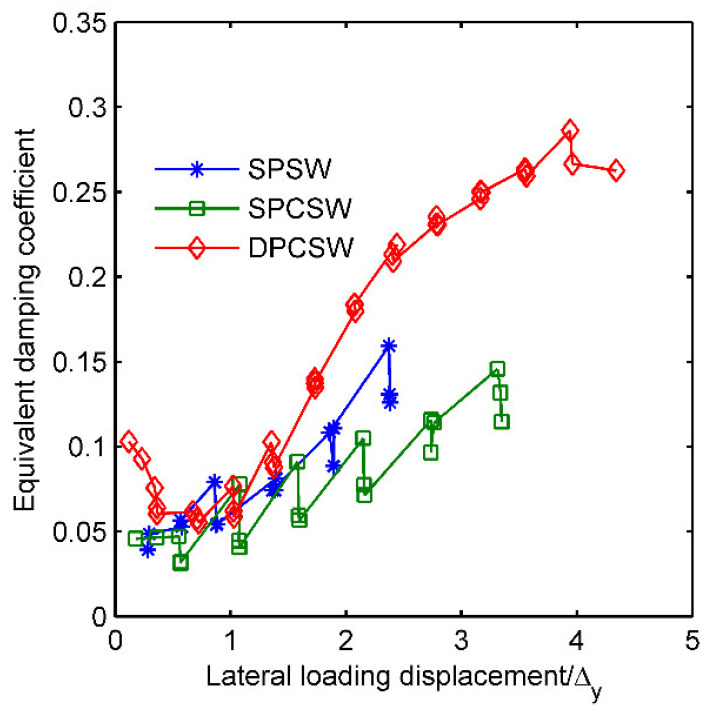
Energy dissipation capacity of specimens.

**Figure 18 materials-15-05737-f018:**
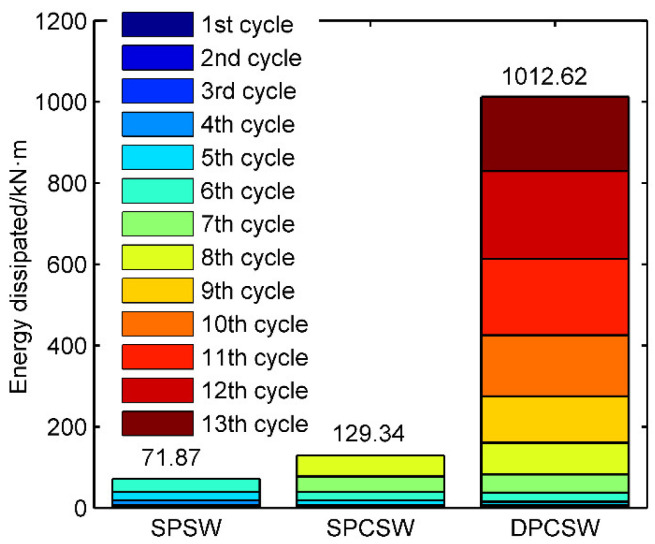
Cumulative energy consumption.

**Figure 19 materials-15-05737-f019:**
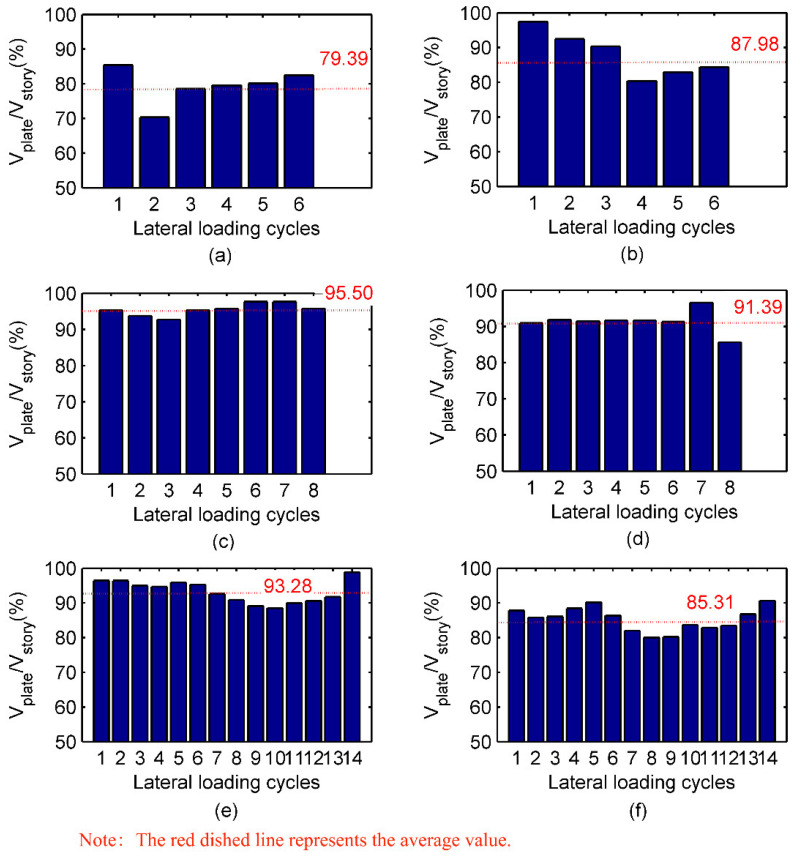
Proportion of shear forces resisted by shear walls: (**a**) the 1st story of specimen SPSW; (**b**) the 2nd story of specimen SPSW; (**c**) the 1st storey of specimen SPCSW; (**d**) the 2nd story of specimen SPCSW; (**e**) the 1st story of specimen DPCSW; (**f**) the 2nd story of specimen DPCSW.

**Figure 20 materials-15-05737-f020:**
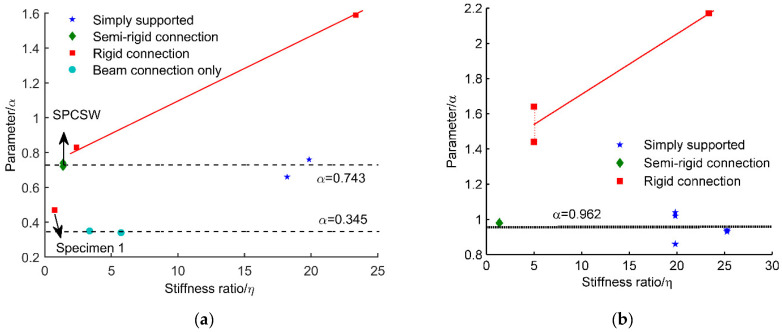
Relationship between parameter η and stiffness ratio α: (**a**) SPSW; (**b**) DPCSW.

**Figure 21 materials-15-05737-f021:**
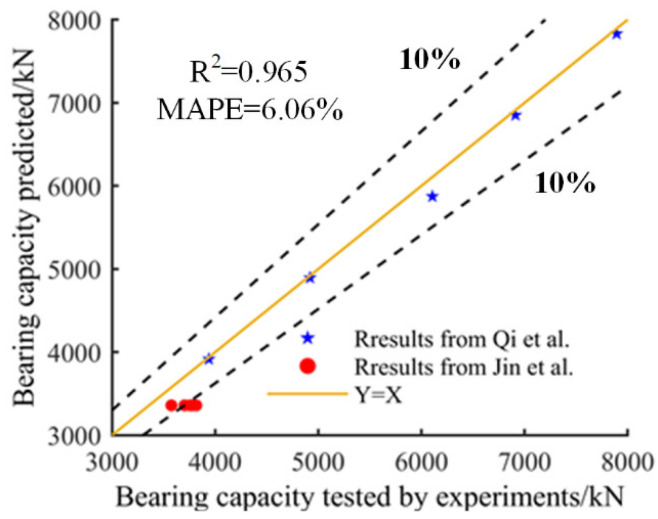
Validation of bearing capacity formula [6,18].

**Table 1 materials-15-05737-t001:** Composition of tested specimens.

Specimens	Steel Frames			Steel Plate		Concrete Panel		Connection Type
	Beams	Columns	Connection Type	Thickness\mm	λ	Type	Thickness/mm	
SPSW	$\begin{array}{l} I\;140\times 80\\ \times 5.5\times 9.1 \end{array}$	HW 125× 125×7×9	Semi-rigid	2	555	——		Welded fish plate
SPCSW				2	555	Single-sided	70 mm	
DPCSW				2	555	Double-sided	35 mm × 2	

Note: λ is the height-thickness ratio of steel plates.

**Table 2 materials-15-05737-t002:** Properties of steel and steel bars.

Positions	Yield Strength, *f*_y_ (MPa)	Ultimate Strength, *f*_u_ (MPa)	*f*_u_/*f*_y_	Elastic Modulus, *E* (MPa)
Frame columns	311	475.5	1.53	2.18 × 10^5^
Frame beams	306	462.5	1.51	
Extended endplate	347	484	1.39	
Steel plates	280	400	1.43	
Steel bars	383	479	1.25	2.05 × 10^5^

**Table 3 materials-15-05737-t003:** Details of characteristic points.

Specimens	Loading Direction	Initial Stiffness	Yield Strength	Yield Drift	Maximum Strength	Drift at Maximum Strength	Failure Strength	Maximum Drift	Failure Stiffness	Ductility
		/kN/mm	/kN	/mm	/kN	/mm	/kN	/mm	/kN/mm	
SPSW	(+)	39.25	362.42	14.47	442.26	27.18	327.45	34.57	9.84	2.39
	(−)	39.74	−367.56	−13.90	−425.58	−26.53	−364.37	−33.57	11.61	2.41
SPCSW	(+)	43.92	396.26	12.72	557.08	34.80	394.16	42.59	10.71	3.35
	(−)	43.86	−348.41	−13.16	−569.66	−33.59	−536.22	−40.80	13.56	3.10
DPCSW	(+)	36.50	508.48	16.50	820.79	58.62	598.46	71.63	11.03	4.34
	(−)	42.27	−486.34	−13.41	−822.67	−46.27	−716.31	−57.42	12.17	4.28

**Table 4 materials-15-05737-t004:** Information of the tested specimens.

		Steel Plate					$t_{c}$/mm	$P_{T}$/kN	$P_{c}/\mathrm{kN}$	$\begin{array}{l} \alpha =\\ P_{T}/P_{c} \end{array}$	Frame Columns				
		t/mm	L/mm	H/mm	$f_{y}$/(N/mm^2^)	λ					Dimensions/mm	$I_{\min}/10^{6}{\mathrm{mm}}^{4}$	$I/10^{6}{\mathrm{mm}}^{4}$	$\eta =I/I_{\min}$	Connection Type
SPSW	SPSW	2	1565	1110	280	555	-	365.49	505.99	0.72	HW125×125×7×9	6.01	8.30	1.38	Semi-rigid
	SPSW-500 [17]	2.2	1100	1100	382.7	500	-	350.44	534.70	0.66	H 216×300×56×16	9.08	165.19	18.20	Simply supported
	SPSW-400 [17]	2.8	1100	1100	324	393	-	440.53	576.15	0.76	H 216×300×56×16	11.6	229.35	19.85	Simply supported
	SPSW-1 [13]	2	1560	1050	283	525	-	813	509.78	1.59	DN 219×4	4.83	112.86	23.36	Rigid
	Specimen 1 [32]	4	1200	1200	356	300	-	465.1	986.58	0.47	HW150×150×7×10	21.4	16.01	0.75	Rigid
	Specimen 2 [33]	3.3	1050	1050	345.27	318	-	575.09	690.72	0.83	HW175×175×7.5×11	11.8	28.16	2.38	Rigid
	Specimen 3 [5]	5	1760	1275	335.7	255	-	602.6	1705.59	0.35	DN 200×8	23.3	78.50	3.37	Beam connected only
	Specimen 4 [5]	5	1760	1275	338.3	255	-	580.4	1718.79	0.34	□ 200×8	23.3	133.33	5.73	Beam connected only
SPSW with single panel	SPCSW	2	1565	1110	280	555	70	372.34	505.99	0.74	HW125×125×7×9	6.01	8.30	1.38	Semi-rigid
SPSW with double panels	DPCSW	2	1565	1110	280	555	35	497.41	505.99	0.98	HW125×125×7×9	6.01	8.30	1.38	Semi-rigid
	BR-SPW1-500 [17]	2.2	1100	1100	382.7	500	50	500.89	534.70	0.94	H 216×300×56×16	9.08	229.35	25.27	Simply supported
	BR-S PW2-500 [17]	2.2	1100	1100	382.7	500	50	499.15	534.70	0.93	H 216×300×56×16	9.08	229.35	25.27	Simply supported
	BR-SPW1-400 [17]	2.8	1100	1100	382.7	393	50	694.05	680.53	1.02	H 216×300×56×16	11.6	229.35	19.85	Simply supported
	BR-SPW2-400 [17]	2.8	1100	1100	382.7	393	50	705.17	680.53	1.04	H 216×300×56×16	11.6	229.35	19.85	Simply supported
	C-SPW-400 [17]	2.8	1100	1100	382.7	393	50	584.61	680.53	0.86	H 216×300×56×16	11.6	229.35	19.85	Simply supported
	Traditional type [15]	4.8	1730	1752	248	365	75	1953	1188.99	1.64	H 333×313×18×28	81.0	405.35	5.00	Rigid
	Innovative type [15]	4.8	1730	1752	248	365	75	1717	1188.99	1.44	H 333×313×18×28	81.0	405.35	5.00	Rigid
	CSPSW-1 [13]	2	1560	1050	230	525	50	899.13	414.31	2.17	DN 219×4	4.83	112.86	23.36	Rigid

Note: t, L, H,
$f_{y}$, λ = thickness, width, height, yield strength and height–thickness ratio of steel plates; $t_{c}$ = thickness of concrete panels; $P_{T}$ = the lateral load corresponding to the yield of the shear walls, which is determined by Figure 8; $P_{c}$ = nominal shear strength of a composite shear wall, which is calculated by $P_{c}=f_{y}Lt/\sqrt{3}$ [34]; $I_{\min}$,I = minimum flexural stiffness and actual flexural stiffness of frame columns, $I_{\min}=0.0031H^{4}t/L$.

## Data Availability

The data were uploaded through FIG files.

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
