# Peer review of "Experimental and Analytical Studies of Prefabricated Composite Steel Shear Walls under Low Reversed Cyclic Loads"

_materials, 2022, doi:10.3390/ma15165737_

Round 1

Reviewer 1 Report

The paper deals with the analysis of the seismic performance of PCSWs with single-sided and double-sided prefabricated RC panels. The research subject is interesting. Several experiments with different sample configurations have been conducted. The results published by other authors have been also analyzed. The paper is well written and structured. However, there are minor issues that should be addressed by the authors before considering the paper for publication. See the comments in the attached pdf file.

Reviewer 2 Report

The work performed by the authors is timely and worthy of investigation. It is one of the most challenging problems in terms of civil structures in order to produce composite concrete with higher structural stability, life, stiffness, and damping parameters. My recommendation is to accept the paper for publication after a minor revision based on the following comments,

*The “Mpa” is erroneous. It shall be “MPa”

*The mathematical relations used from literature shall be cited with the proper references.

*Residual stresses resulting from manufacturing are important for such a structure. Has it been considered in the analysis?

*Authors could provide a FEM analysis e.g., with Abaqus for validation purposes.
